# TOWARDS ZERO MEMORY FOOTPRINT SPIKING NEURAL NETWORK TRAINING

## ABSTRACT

Spiking Neural Networks (SNNs), as representative brain-inspired neural networks, emulate the intrinsic characteristics and functional principles of the biological brain. With their unique structure reflecting biological signal transmission through spikes, they have achieved significant success in processing temporal data. However, the training of SNNs demands a substantial memory footprint due to the added storage needs for spikes or events, resulting in intricate architectures and dynamic configurations. In this paper, to address memory constraints in SNN training, we introduce an innovative framework characterized by a remarkably low memory footprint. We **(i)** design a reversible spiking neuron that retains a high level of accuracy. Our design is able to achieve a $58.65\times$ reduction in memory usage compared to the current spiking neuron. We **(ii)** propose a unique algorithm to streamline the backpropagation process of our reversible spiking neuron. This significantly trims the backward Floating Point Operations Per Second (FLOPs), thereby accelerating the training process in comparison to the current reversible layer backpropagation method. By using our algorithm, the training time is able to be curtailed by $23.8\%$ relative to existing reversible layer architectures.

## 1 INTRODUCTION

Spiking Neural Networks (SNNs) have gained significant recognition in the realm of bio-inspired neuromorphic computing. In contrast to traditional Deep Neural Networks (DNNs), SNNs possess a unique mechanism that processes information across multiple timesteps and impulse events, commonly referred to as spikes (Davies et al., 2018; Viale et al., 2021). This inherent ability for temporal processing enables SNNs to excel in tasks requiring real-time or sequential data interpretation. An illustrative example of this prowess is observed in robot navigation tasks utilizing Intel's Loihi platform (Davies et al., 2018), underscoring SNNs' proficiency in managing temporal data. Further, works such as (Kim & Panda, 2021) emphasize the advantages of SNNs over DNNs when handling sparse datasets, exemplified by data from dynamic vision sensors (DVS). These insights underscore the potential of SNNs across diverse applications where processing sequential or time-varying signals is crucial.

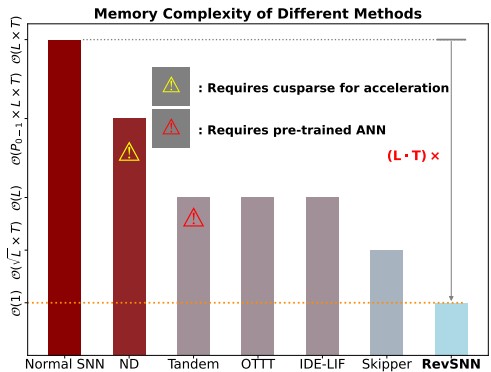

Figure 1: Comparison of memory complexity between our RevSNN and other current SOTA Memory-Efficient SNN Training Techniques.

Despite their numerous advantages, one major bottleneck in deploying SNNs is their memory consumption. For a DNN of depth $L$, the memory complexity is $\mathcal{O}(L)$. However, an SNN of equivalent depth incorporates multiple timesteps $T$ in its computation, amplifying its memory complexity to $\mathcal{O}(L \times T)$. To illustrate, while the memory demand during the training of a DNN like ResNet19 is a mere 0.6 GB, an SNN with the same architecture surges to 12.34 GB (~20 ×)

with a timestep of 10. Such heightened memory requirements pose significant challenges for SNN integration into resource-limited environments, notably in IoT-Edge devices (Putra & Shafique, 2021).

To tackle the SNN's memory consumption challenge, various methods have been proposed. Shown in Fig. 1, the ND method by Huang et al. (2023), utilizing sparse training, reduces memory demands from $\mathcal{O}(L \times T)$ in the original SNN to $p \times \mathcal{O}(L \times T)$, where $0 \leq p \leq 1$. However, this approach necessitates specialized hardware support, such as the CuSparse library. Works such as Tandem (Wu et al., 2021a), OTTT (Xiao et al., 2022), and IDE (Xiao et al., 2021) further reduce memory requirements to $\mathcal{O}(L)$, with Tandem needing pre-trained Artificial Neural Network. Leveraging the checkpoint techniques, Skipper (Singh et al., 2022) achieves a complexity of $\mathcal{O}(\sqrt{L} \times T)$. Overall, these methods lack scalability due to limited memory reduction when SNN layers and timesteps increase.

In this paper, we address the following question: *Does a scalable training memory reduction method exist that remains scalable irrespective of increased layers and timesteps? If so, how can such a training method be designed?* To that end, we present a novel reversible spiking neuron that substantially lowers the memory footprint . Our contributions are summarized as follows:

- We recalculate all the intermediate states on-the-fly, rather than storing them during the backward propagation process. Notably, in comparison, our method realizes a memory complexity of $\mathcal{O}(1)$, shown in Fig. 1. In other words, our training memory reduction method is scalable when increasing SNN layers and timesteps.
- To enhance the efficiency, we design a reverse computation graph for the backpropagation process of our reversible spiking neuron, eliminating the need to rebuild the forward computation graph, which significantly reduces the training time compared with the original reversible layer backpropagation method.
- Empirical evaluation of vast datasets shows that our method could retains the same level of accuracy during training process, compared to state-of-the-art (SOTA) methods.

Experimental results show that our approach markedly surpasses the SOTA **Memory-Efficient** SNN training with reductions of $8.01\times$, $9.51\times$, and $3.93\times$ on the CIFAR10, CIFAR100, and DVS-CIFAR10 datasets respectively. Incorporating our reversible spiking neurons into the OTTT method for the DVS128-Gesture dataset, we achieve a notable $1.34\times$ reduction compared to the original OTTT, maintaining high accuracy levels. Moreover, our method reduces the FLOPs needed for the backpropagation by a factor of 23% compared to the existing reversible layer backpropagation method, thus accelerating the training process. We hope these advances would pave the way for more efficient and scalable SNN implementations, enabling the deployment of these biologically inspired networks across a wider range of applications and hardware platforms.

## 2 BACKGROUND AND RELATED WORKS

### 2.1 SPIKING NEURAL NETWORK

The cardinal features that distinguish SNNs from conventional neural networks include: **(i)** Their inherent operation over multiple timesteps, emulating the temporal dynamics of information processing found in biological systems. This attribute broadens their ability to capture and interpret time-dependent patterns and sequences (Ghosh-Dastidar & Adeli, 2009). **(ii)** A unique mechanism of data handling through *spikes*. Unlike traditional networks that process continuous values, SNNs convey information via these discrete-time events. This spiking mechanism offers a more biologically faithful representation of neuronal signaling and emphasizes their potential to emulate the genuine communication patterns of neurons in the human brain (Tavanaei et al., 2019; Koravuna et al., 2023).

There are several spiking neural models in the literature: Leaky Integrate and Fire (LIF) (Dayan & Abbott, 2005), Hodgkin-Huxley (HH) (Hodgkin & Huxley, 1952), and Izhikevich (Izhikevich, 2003). The LIF model is the most commonly utilized, and our reversible spiking neuron in this paper is constructed based on this model.

The LIF model's core involves two primary phases: - **Integration**: Signals to the neuron accumulate over time. However, the LIF neuron, unlike a perfect integrator, has a *leaky* attribute, leading to the decay of the neuron's accumulated voltage towards its resting state without new inputs. -

**Firing**: When the integrated voltage surpasses a set threshold, the neuron releases a spike. Following the firing, the voltage resets, generally to a value beneath the threshold, and the procedure begins anew. The specific firing function often uses the Heaviside step function (Wu et al., 2021a;b) or its derivatives (Meng et al., 2022; Nicola & Clopath, 2017).

In conclusion, SNNs have found practical applications across various fields. Specifically, they have made significant strides in areas including segmentation (Kim et al., 2022; Patel et al., 2021) and detection (Kim et al., 2020). In the biomedical domain, SNNs have been extensively explored for tasks such as MRI image segmentation (Ahmadi et al., 2021) and ECG classification (Yan et al., 2021). Their biologically inspired architecture and unique data processing capabilities position SNNs as a powerful tool, bridging the gap between computational neuroscience and real-world applications. As advancements continue, the scope and impact of SNNs are poised to grow even further.

## 2.2 EXISTING MEMORY-EFFICIENT TECHNIQUES IN SNN TRAINING

Training SNNs can be computationally intensive, often demanding significant memory resources. Given the intrinsic temporal characteristics of SNNs, training them involves processing information over several timesteps, which further amplifies the memory requirements (Bauer et al., 2023). This has spurred research into developing memory-efficient techniques tailored for SNN training.

Just as with conventional networks, SNNs can also adopt some traditional memory-saving techniques, such as checkpointing and sparse training. The work by Singh et al. (2022) applied checkpointing to SNNs and, compared to the baseline SNN-BPTT, achieved a reduction in memory usage ranging from $3.3\times$ to $8.4\times$, with an average of $6.7\times$. Additionally, the study by Huang et al. (2023) utilized sparse training for SNNs, and the results revealed that the training cost of NDSNN is merely $40.89\%$ of the LTH training cost when implemented on ResNet-19.

One of the primary reasons for the substantial memory consumption in SNNs is the need to retain computational graphs for multiple time steps during the backpropagation process. This has led to the development of techniques that focus on optimizing the backpropagation process in SNNs to conserve memory. An exemplar is (Xiao et al., 2022), which compressed the multi-time step backpropagation into a single time step, resulting in significant memory savings. When the timestep is set to six, this approach can reduce memory consumption by approximately 2 to 3 times.

## 3 REVERSIBLE SPIKING NEURON

### 3.1 TRAINING MEMORY ANALYSIS

During the training process of spiking neural networks, the activation values occupy the main memory storage space. The activation value memory analysis schematic diagram is shown in Fig. 2.

In this figure, we use the VGG-13 architecture (Simonyan & Zisserman, 2014) with ten timesteps as an example. The percentage values represent the memory footprint ratio of each part in the entire network. The left diagram is the original SNN where the activation values of spikes account for 90.9% of the memory usage, and the output potentials of each neuron occupy 9.1% of the memory. The right diagram is our designed reversible SNN, which only requires saving the output potentials of each neuron, without storing all intermediate values, thus significantly saving memory. The intermediate activation values will be regained during the backpropagation process through our inverse calculation equation. In this example, our method is able to save 90.9% of the memory used for activation values. The exact amount of memory saved by our method is shown in Section 5.2.

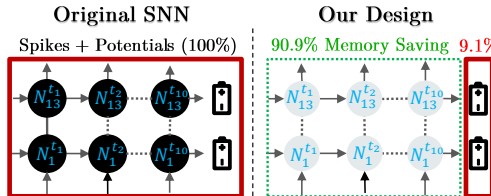

Figure 2: Memory comparison between the activation value of the original SNN network and our reversible SNN network. ▭: Activation value bound to memory storage; ⬚: Activation value free from memory storage; ●: Original spiking neuron; ○: Our reversible spiking neuron; ▭▶: Output potential of the spiking neuron; $N_i^{t_j}$: spiking neuron on layer $i$ timesteps $j$.

## 3.2 REVERSIBLE SPIKING NEURON FORWARD CALCULATION

Our forward algorithm is in the upper section of Fig. 3. ❶: The various input states $\mathcal{S} = (\mathcal{X}, \mathcal{V})$ of

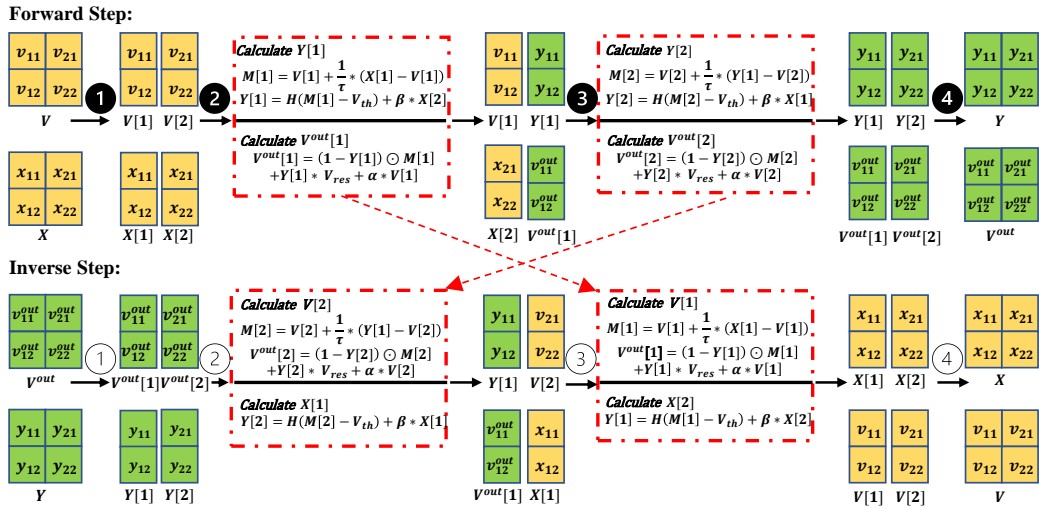

Figure 3: This reversibility demo use $2 \times 2$ Input as an illustrative example and shows our forward and inverse calculations. $- - - \blacktriangleright$ : The origin of the equations in the inverse process.

each neuron are evenly divided into two groups along the last dimension. Namely: $\mathcal{S} = [\mathcal{S}_1, \mathcal{S}_2]$.

❷: Calculate the first part of output $\mathcal{M}_1^t$ and $\mathcal{Y}_1$:

$$\mathcal{M}_1^t = \mathcal{V}_1^{t-1} + \frac{1}{\tau} \cdot \left( \mathcal{X}_1^t - \mathcal{V}_1^{t-1} \right) \quad (1) \qquad \mathcal{Y}_1^t = \mathcal{H} \left( \mathcal{M}_1^t - V_{th} \right) + \beta \cdot \mathcal{X}_2^t \quad (2)$$

$\mathcal{M}_1^t$ is the membrane potential of the first half neuron at time $t$. $\mathcal{V}_2^{t-1}$ is the input potential of the second half neuron at time $t - 1$. $\tau$ is the time constant. $\mathcal{X}_2^t$ is the input to the second half neuron at time $t$. $V_{th}$ is the threshold voltage of the neurons. $\mathcal{H}()$ is the Heaviside step function. $\beta$ is a scaling factor for the input. $\beta \cdot \mathcal{X}_2^t$ will help $\mathcal{Y}_1^t$ to collect information about the second half of the input in the next step. $\mathcal{M}, \mathcal{V}, \mathcal{X}, \mathcal{Y} \in \mathbb{R}^{\prod_{i=1}^{n} d_i}$, $V_{th} \in \mathbb{R}$. Then we calculate the first part of the output voltage:

$$\mathcal{V}_1^t = \left( 1 - \mathcal{Y}_1^t \right) \odot \mathcal{M}_1^t + \mathcal{Y}_1^t \cdot V_{res} + \alpha \cdot \mathcal{V}_1^{t-1} \quad (3)$$

$\mathcal{V}_1^t$ is the output potential of the first half neuron at time $t$. $V_{res}$ is the reset voltage of the neurons. $\alpha$ is a scaling factor for the membrane potential.

❸: Use the first part of output $\mathcal{Y}_1$ to calculate the second part $\mathcal{M}_2^t$ and $\mathcal{Y}_2$:

$$\mathcal{M}_2^t = \mathcal{V}_2^{t-1} + \frac{1}{\tau} \left( \mathcal{Y}_1^t - \mathcal{V}_2^{t-1} \right) \quad (4) \qquad \mathcal{Y}_2^t = \mathcal{H} \left( \mathcal{M}_2^t - V_{th} \right) + \beta \cdot \mathcal{X}_1^t \quad (5)$$

$\mathcal{M}_2^t$ is the membrane potential of the second half neuron at time $t$. $\mathcal{Y}_2^t$ is the output of the second half neuron at time $t$. We calculate the second part of the output voltage by:

$$\mathcal{V}_2^t = \left( 1 - \mathcal{Y}_2^t \right) \odot \mathcal{M}_2^t + \mathcal{Y}_2^t \cdot V_{res} + \alpha \cdot \mathcal{V}_2^{t-1} \quad (6)$$

$\mathcal{V}_2^t$ is the output potential of the second half neuron at time $t$, $V_{res} \in \mathbb{R}$.

❹: For all the output states $\mathcal{S}_{output} = ([\mathcal{Y}_1, \mathcal{Y}_2], [\mathcal{V}_1^t, \mathcal{V}_2^t])$, combine them by the last dimension.

## 3.3 REVERSIBLE SPIKING NEURON INVERSE CALCULATION

The purpose of the inverse calculation is to use the output results to obtain the unsaved input values. i.e. Use $\mathcal{Y}$ and $\mathcal{V}_{output}$ to calculate $\mathcal{X}$ and $\mathcal{V}$. Our inverse algorithm is in the lower section of Fig. 3. ①:

For all the output states $\mathcal{S}_{output} = (\mathcal{Y}, \mathcal{V}_{output})$, divide them into two groups by the last dimension in the same way as in the first step of forward calculation, namely: $\mathcal{S}_{output} = [\mathcal{S}_{output}1; \mathcal{S}_{output}2]$

②: Calculate $\mathcal{V}_2^{t-1}$ by combine Eq. (4) and calculate $\mathcal{X}_1^t$ by combine Eq. (4), (5), and (6), simplify:

$$\mathcal{V}_2^{t-1} = \frac{\mathcal{V}_2^t - (1 - \mathcal{Y}_2) \cdot \frac{1}{\tau} \odot \mathcal{Y}_1 - \mathcal{Y}_2 \cdot V_{reset}}{(1 - \mathcal{Y}_2) \cdot (1 - \frac{1}{\tau}) + \alpha} \quad (7) \qquad \mathcal{X}_1^t = \frac{\mathcal{Y}_2^t - \mathcal{H}(\mathcal{M}_2^t - V_{th})}{\beta} \quad (8)$$

③: Calculate $\mathcal{V}_1^{t-1}$ by combine Eq. (1) and calculate $\mathcal{X}_2^t$ by combine Eq. (1), (2) and (3), simplify:

$$\mathcal{V}_1^{t-1} = \frac{\mathcal{V}_1^t - (1 - \mathcal{Y}_1) \cdot \frac{1}{\tau} \odot \mathcal{X}_1^t - \mathcal{Y}_1 \cdot V_{reset}}{(1 - \mathcal{Y}_1) \cdot (1 - \frac{1}{\tau}) + \alpha} \quad (9) \qquad \mathcal{X}_2^t = \frac{\mathcal{Y}_1^t - \mathcal{H}(\mathcal{M}_1^t - V_{th})}{\beta} \quad (10)$$

④: For all the input states $\mathcal{S} = ([\mathcal{X}_1, \mathcal{X}_2], [\mathcal{V}_1^{t-1}, \mathcal{V}_2^{t-1}])$, combine them by the last dimension.

## 4  INVERSE GRADIENT CALCULATION

While our reversible architecture markedly reduces memory consumption, it does introduce computational overhead due to two main factors: (i) The need to recompute previously unstored activation values, and (ii) Many past reversible layers borrowed the backpropagation technique from checkpointing (THUDM, 2023; Fan et al., 2020). This approach recalculates intermediate activations to reconstruct a forward computational graph for gradient derivation, adding computational overhead and increasing total computation time. This design is unnecessary in the reversible architecture. This scenario is prevalent across all existing architectures of reversible layers, including Reversible GNN (Li et al., 2021a), Reversible CNN (Gomez et al., 2017), and so on. To reduce the training time, we have designed a new algorithm called the inverse gradient calculation method, which can substantially decrease the number of FLOPs during the backpropagation process compared to the original reversible architecture. Our design is shown in Fig. 4.

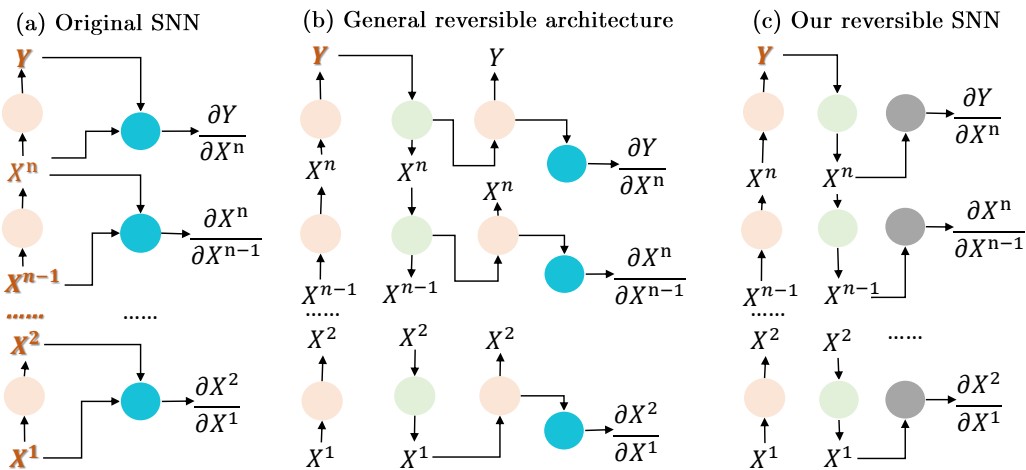

Figure 4: Three different architectures for comparison. 🔴: Forward function, 🟢: inverse function, 🔵: $\frac{\partial \mathcal{X}^n}{\partial \mathcal{X}^{n-1}}$ derivative, ⚫:**Part of** $\frac{\partial \mathcal{X}^{n-1}}{\partial \mathcal{X}^n}$ derivative, **Brown** variables: Cached values.

The left diagram illustrates the original forward and backward processes. The middle diagram depicts the original calculation process for reversible layers. It contains four steps:

1. The input $\mathcal{X}$ pass the forward function to compute the output $\mathcal{Y}$, without storing the input data to conserve memory.
2. For each layer $n$: The output $\mathcal{X}^n$ of this layer passes the inverse function to compute the input $\mathcal{X}^{n-1}$ of this layer. This process starts with the final output $\mathcal{Y}$.

3. For each layer $n$: The input $\boldsymbol{\mathcal{X}}^{n-1}$ passes through the forward function again to reconstruct the forward computational graph, which facilitates gradient computation.
4. For each layer $n$: Compute the gradient $\frac{\partial \boldsymbol{\mathcal{X}}^n}{\partial \boldsymbol{\mathcal{X}}^{n-1}}$ based on the forward computational graph.

The right diagram is our design with three steps:

1. The input $\boldsymbol{\mathcal{X}}$ pass the forward function to compute the output $\boldsymbol{\mathcal{Y}}$, without storing the input data to conserve memory.
2. For each layer $n$: The output $\boldsymbol{\mathcal{X}}^n$ of this layer passes the inverse function to compute the input $\boldsymbol{\mathcal{X}}^{n-1}$ of this layer and construct an inverse computational graph.
3. For each layer $n$: Compute the gradient $\frac{\partial \boldsymbol{\mathcal{X}}^n}{\partial \boldsymbol{\mathcal{X}}^{n-1}}$ based on the inverse computational graph.

Below is the specific calculation formula of the $\frac{\partial \mathbf{X}^n}{\partial \mathbf{X}^{n-1}}$ based on the inverse computation graph, and the derivation process is in the Appendix.

$$\frac{\partial \boldsymbol{\mathcal{X}}^n}{\partial \boldsymbol{\mathcal{X}}_1^{n-1}} = \frac{\theta}{2 + (\pi \cdot \theta \cdot (\boldsymbol{\mathcal{M}}_1^t - V_{th}))^2} \cdot \frac{1}{\tau} \odot \left(1 + \frac{\theta}{2 + (\pi \cdot \theta \cdot (\boldsymbol{\mathcal{M}}_2^t - V_{th}))^2} \cdot \frac{1}{\tau}\right) + \beta \quad (11)$$

$$\frac{\partial \boldsymbol{\mathcal{X}}^n}{\partial \boldsymbol{\mathcal{X}}_2^{n-1}} = \frac{\theta}{2 + (\pi \cdot \theta \cdot (\boldsymbol{\mathcal{M}}_2^t - V_{th}))^2} + \beta \quad (12)$$

All the variables in Eq. (11) and Eq. (12) have the same meaning as the variables in Eq. (1) - Eq. (10) and $\theta$ is an adjustable constant parameter.

The ability to perform computational graph inverse computation in our algorithm is based on that our forward function has symmetry with the inverse computation function.

For the original reversible network:

$$FLOPS_{backward}^{ori} = FLOPS_{inverse} + FLOPS_{forward} + FLOPS_{\frac{\partial \boldsymbol{\mathcal{X}}^n}{\partial \boldsymbol{\mathcal{X}}^{n-1}}} \quad (13)$$

For our reversible network:

$$FLOPS_{backward}^{our} = FLOPS_{inverse} + FLOPS_{part\ of\ \frac{\partial \boldsymbol{\mathcal{X}}^{n-1}}{\partial \boldsymbol{\mathcal{X}}^n}} \quad (14)$$

Compared to the standard reversible network, our method reduces FLOPS by 23%. The FLOPS analysis is shown in the Appendix and the detailed time mearsurement is shown in the Section 5.3.

## 5 EXPERIMENT

We first benchmarked our design against SOTA SNN training methods on multiple datasets, and then integrated our reversible spiking neuron into various architectures. Our primary aims are to highlight the memory efficiency of our method over the conventional spiking neuron and demonstrate the speed benefits of our backpropagation design compared to the existing reversible backpropagation method. An ablation study was also conducted to assess different parameters' effects and the influence of input group divisions on our model's performance.

Experiments ran on an RTX6000 GPU using PyTorch 1.13.1 and CUDA 11.4. We verified the consistency of inverse and forward calculations using `torch.allclose(rtol=1e^{-06}, atol =1e^{-10})`, achieving accurate results. Hyperparameters are detailed in the Appendix.

### 5.1 COMPARISON WITH THE SOTA METHODS

We compared our approach with the current SOTA methods in **Memory Efficiency** during the SNN training process across two standard image classification datasets, CIFAR10 and CIFAR100, as well as two neuromorphic datasets, DVS-CIFAR10 and DVS128gesture. The results are shown in Table 1.

Table 1: Comparison of our work with the SOTA methods in **Memory Efficiency** at SNN training phase. For all the works: Batch size = 128. †: We conducted experiments using provided open-source code when available. *: If not, the results were generated with our own implementation.

| Dataset | Method | Architecture | Time-steps | Accuracy | Memory(GiB) |
|---|---|---|---|---|---|
| CIFAR10 | OTTT (Xiao et al., 2022) | VGG(sWS) | 6 | 93.52% | 4 |
| | S2A-STSU Tang et al. (2022) | ResNet-17 | 5 | 92.75% | 27.93 |
| | IDE-LIF (Xiao et al., 2021) | CIFARNet-F | 30 | 91.74% | 2.8 |
| | Hybrid (Rathi et al., 2020) | VGG-16 | 100 | 91.13% | 9.36 |
| | Tandem (Wu et al., 2021a) | CifarNet | 8 | 89.04% | 4.2 |
| | Skipper (Singh et al., 2022) | VGG-5 | 100 | 87.44% | 4.6 |
| | **RevSNN(Ours)** | ResNet-18 | 4 | 91.87% | **1.101 ↓ 8.01× (Avg.)** |
| CIFAR100 | IDE-LIF† (Xiao et al., 2021) | CIFARNet-F | 30 | 71.56% | 2.95† (Maqing, 2021) |
| | OTTT (Xiao et al., 2022) | VGG(sWS) | 6 | 71.05% | 4.04 |
| | S2A-STSU (Tang et al., 2022) | VGG-13 | 4 | 68.96% | 31.05 |
| | Skipper (Singh et al., 2022) | VGG-5 | 100 | 66.48% | 4.6 |
| | **RevSNN(Ours)** | ResNet-18 | 4 | 71.13% | **1.12 ↓ 9.51× (Avg.)** |
| DVS-CIFAR10 | STBP-tdBN (Zheng et al., 2021) | ResNet-19 | 10 | 67.8% | 11.5† (ThiswinEx, 2021) |
| | Tandem (Wu et al., 2021a) | CifarNet | 8 | 65.59% | 6.79† (DeepSpike, 2021) |
| | Rollout (Kugele et al., 2020) | DenseNet | 10 | 66.8% | 15.3* |
| | BPTT (Fang et al., 2021) | 7-layer CNN | 20 | 74.8% | 27.95† (Fangwei, 2021) |
| | **RevSNN(Ours)** | VGG-16 | 20 | 72.11% | **3.91 ↓ 3.93× (Avg.)** |
| DVS128-Gesture | BPTT (Fang et al., 2021) | 8-layerCNN | 20 | 96.88% | 137.10† (Fangwei, 2021) |
| | SLAYER (Shrestha & Orchard, 2018) | 8-layerCNN | 300 | 93.64% | 5.18† (Sumit, 2018) |
| | DECOLLE (Kaiser et al., 2020) | 3-layerCNN | 1800 | 95.54% | 5.03† (Lab, 2020) |
| | OTTT (Xiao et al., 2022) | VGG(sWS) | 20 | 96.88% | 28.44† (Xiao, 2022) |
| | **RevOTTT(Ours)** | VGG(sWS) | 20 | 96.75% | **21.16 ↓ 1.34×** |

We subsequently applied our reversible spiking neuron to the current SOTA techniques in terms of SNN **Accuracy** and compared it with the original methods. The results are shown in Table 2.

Table 2: Comparison of our work with the SOTA methods in terms of SNN **Accuracy**. For all the works: Batch size = 128. †: We conducted experiments using provided open-source code when available. *: If not, the results were generated with our own implementation.

| Dataset | Method | Architecture | Time-steps | Accuracy | Memory(GiB) |
|---|---|---|---|---|---|
| CIFAR10 | Dspike (Li et al., 2021b) | ResNet-18 | 6 | 94.25% | 5.78* |
| | **RevDspike(Ours)** | ResNet-18 | 6 | 93.43% | **2.14 ↓ 2.70×** |
| | DSR (Meng et al., 2022) | PreAct-ResNet-18 | 20 | 95.40% | 25.11† (Meng, 2022) |
| | **RevDSR(Ours)** | PreAct-ResNet-18 | 20 | 95.35% | **5.73 ↓ 4.38×** |
| CIFAR100 | Dspike (Li et al., 2021b) | ResNet-18 | 6 | 74.24% | 5.78* |
| | **RevDspike(Ours)** | ResNet-18 | 6 | 73.28% | **2.14 ↓ 2.70×** |
| | DSR (Meng et al., 2022) | PreAct-ResNet-18 | 20 | 78.50% | 25.11† (Meng, 2022) |
| | **RevDSR(Ours)** | PreAct-ResNet-18 | 20 | 78.21% | **5.73 ↓ 4.38×** |
| Tiny-ImageNet | ND(Dense) (Huang et al., 2023) | VGG-16 | 5 | 39.45% | 3.99 |
| | ND(90% Sparsity) (Huang et al., 2023) | VGG-16 | 5 | 39.12% | 3.78 |
| | ND(99% sparsity) (Huang et al., 2023) | VGG-16 | 5 | 33.84% | 3.76 |
| | **RevND(Ours)** | VGG-16 | 5 | 39.73% | **2.01 ↓ 1.99×** |
| | ND(Dense) (Huang et al., 2023) | ResNet-19 | 5 | 50.32% | 5.29 |
| | ND(90% Sparsity) (Huang et al., 2023) | ResNet-19 | 5 | 49.25% | 5.11 |
| | ND(99% sparsity) (Huang et al., 2023) | ResNet-19 | 5 | 41.96% | 5.09 |
| | **RevND(Ours)** | ResNet-19 | 5 | 50.63% | **2.47 ↓ 2.14×** |

Compared to the SOTA **Memory-Efficient** SNN training, our approach (RevSNN) significantly achieves a $8.01\times$ reduction on the CIFAR10 dataset; a $9.51\times$ reduction on the CIFAR100 dataset; and a $3.93\times$ reduction on the DVS-CIFAR10 dataset on average. To further evaluate the versatility of our reversible spiking neurons, we incorporated them into the OTTT method (RevOTTT) for the DVS128-Gesture dataset. The results are compelling: a $1.34\times$ reduction compared to the original OTTT approach, all while preserving a high degree of accuracy. Against **Accuracy-Driven** SNN training, our spiking neuron integrated into SOTA methods (RevDespike, RevDSR, RevND) yielded substantial memory savings: $2.70\times$ for Dspike and $4.38\times$ for DSR on CIFAR datasets. On Tiny-ImageNet, using our neuron with ND method's VGG-16 and ResNet-19 architectures resulted in $1.99\times$ and $2.14\times$ reductions, respectively, with accuracy surpassing the original Dense model.

## 5.2 MEMORY CONSUMPTION EVALUATION

We explored the memory savings of our reversible spiking neuron by incorporating it into various architectures, including VGG (11, 13, 16, 19) and ResNet (19, 34, 50, 101), using the CIFAR-10

dataset with a batch size of 128. For VGG architectures, we analyzed memory usage over 1 to 20 timesteps, while for ResNet, it was over 1 to 10 timesteps. The findings are shown in Fig. 5. Notably, with the VGG-19 architecture at 20 timesteps, the memory usage for our reversible spiking neuron remains under 200MB, in stark contrast to the 9032MB required using conventional spiking neuron. For ResNet-101 at 10 timesteps, the comparison is 1382MB to 28993MB. As we scale model layers and timesteps, the memory efficiency of our reversible spiking neuron is even more evident. For instance, VGG-19 at 20 timesteps sees a $58.65\times$ memory reduction. Detailed data are shown in the Appendix.

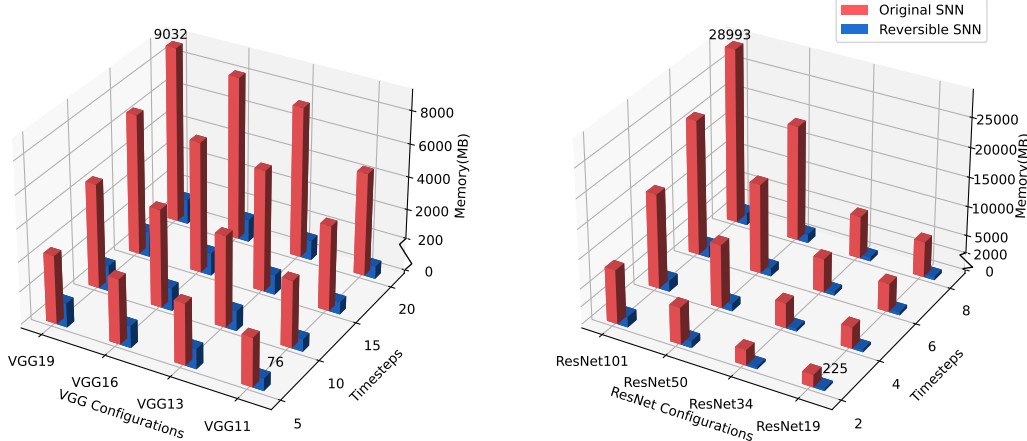

Figure 5: Memory comparison between normal spiking neuron and our reversible spiking neuron.

These experimental results align with our theoretical analysis in Section 3.1, further validating that our design is able to significantly reduce memory usage.

## 5.3 TRAINING TIME EVALUATION

To compare the efficiency of our backpropagation design with the traditional reversible method, we evaluated two backpropagation architectures for our reversible spiking neuron: one with the conventional method and another with our design. We used VGG architectures (VGG-11 to VGG-19) over timesteps from 1 to 10 and compared the training iteration times on CIFAR-10 datasets for three scenarios: original spiking neuron, reversible spiking neuron with conventional backpropagation, and reversible spiking neuron with our method. All tests were conducted on an RTX6000 GPU with a batch size of 64.

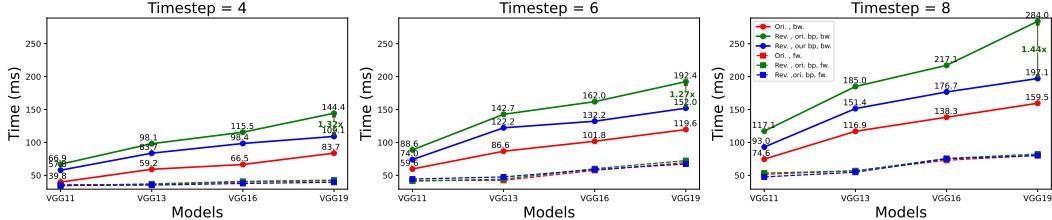

Figure 6: Training time analysis. Solid lines: Backward process's duration; Dashed lines: Forward process's duration; Red lines: Training time for the original SNN; Green lines: Training time for the reversible SNN using the original reversible layer backpropagation method; Blue lines: Training time for the reversible SNN employing our proposed backpropagation architecture.

Fig. 6 presents our measurement of the training time when the number of timesteps is set to 4, 6, and 8. Forward computation times across the three methods are comparable. The original spiking neuron boasts the quickest backward time as it stores all intermediate values, avoiding recalculations. Among reversible spiking neurons, our design speeds up the backward process by $20\% - 30\%$ compared to

the traditional reversible method. This advantage grows with larger networks; for instance, under VGG-19 at 8 timesteps, our method saves $\mathbf{23.8}\%$ of total training time. These findings match our theoretical predictions in Section 4. Further data is in the Appendix.

### 5.4 ABLATION STUDY

**Effects of parameters $\alpha$ and $\beta$ in our equations**

In Eq. (2) and Eq. (3), we have two parameters: $\alpha$ and $\beta$. The optimal setting for the parameter $\beta$ is 1, as this maximizes the preservation of the original features of the data. We conduct experiments to assess the impact of the $\alpha$ parameter on the model's performance. We vary the $\alpha$ parameter from 0.05 to 0.8, and then employ architectures VGG-19, VGG-16, VGG-13, and VGG-11 to evaluate the accuracy on the CIFAR100 dataset. The results are shown on the left of Fig. 7. We observe that varying $\alpha$ within the range of 0.05 to 0.8 impacts the final accuracy by approximately 1%. Generally, the model exhibits optimal performance when $\alpha$ is set between 0.1 to 0.2.

**Effects of number of groups for the various states**

In Section 3.2, we propose splitting input states into two groups along the last dimension. However, this poses problems if the tensor's last dimension is odd. To solve this, we adapt the original algorithm to divide inputs based on the last dimension's element count $n$. This sequential processing with Eq. (1) - (3) for each group enhances our algorithm's flexibility. To assess the number of groups' impact, we adjusted some fully connected layers in ResNet-19, ResNet-18, VGG-16, and VGG-13 networks from 128 to 144 activations for varied factor possibilities. We tested performance on CIFAR100 with groups ranging from 2 to 144, shown in Fig. 7. Results suggest More groups enhance accuracy, often surpassing the original spiking neuron due to improved data representation.

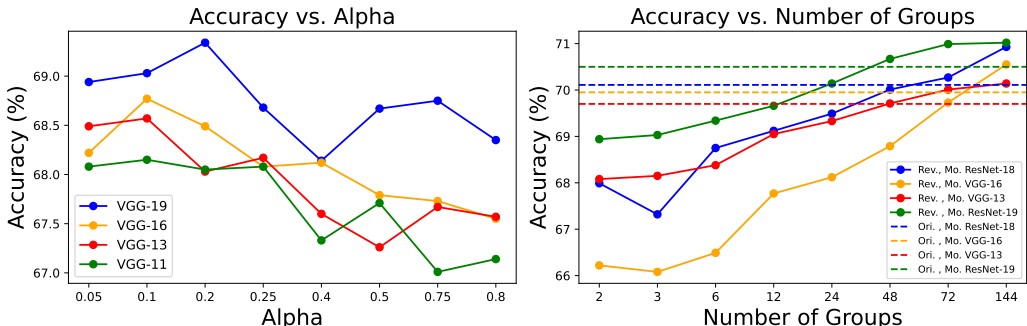

Figure 7: **Left Figure**: Test VGG-19,VGG-16,VGG-13,VGG-11 models on CIFAR100 dataset by using different $\alpha$ settings. **Right Figure**: Change activations number from 128 to 144 for some fully connected layers inside ResNet-19, ResNet-18, VGG-16, VGG-13 and test model performance for different numbers of groups on CIFAR100. Rev.: Reversible spiking neuron. Ori.: Original spiking neuron. Mo.: Modified network (Change some fully connected layers).

## 6  CONCLUSION AND DISCUSSION

This work addresses a fundamental bottleneck of current deep SNNs: their high GPU memory consumption. We have designed a novel reversible spiking neuron that is able to reduce memory complexity from $\mathcal{O}(n^2)$ to $\mathcal{O}(1)$. Specifically, our reversible spiking neuron allows our SNN network to achieve $\mathbf{8.01}\times$ greater memory efficiency than the current SOTA SNN memory-efficient work on the CIFAR10 dataset, and $\mathbf{9.51}\times$ greater on the CIFAR100 dataset on average. Furthermore, in order to tackle the prolonged training time issue caused by the need for recalculating intermediate values during backpropagation within our designed reversible spiking neuron, we have innovated a new backpropagation approach specifically suited for reversible architectures. This innovative method, when compared to the original reversible layer architecture, achieves a substantial reduction in overall training time by $\mathbf{23.8}\%$. As a result, we are able to train over-parameterized networks that significantly outperform current models on standard benchmarks while consuming less memory.

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
