# OpenReview forum: "Towards Zero Memory Footprint Spiking Neural Network Training"
_ICLR.cc/2024/Conference — Submitted to ICLR 2024_

### Official Review · Reviewer_j4UM · 2023-10-27

**Soundness:** 2 fair
**Presentation:** 2 fair
**Contribution:** 2 fair
**Rating:** 5
**Confidence:** 5

**Summary:**

Training normal spiking neural networks (SNNs) directly incurs a significant memory cost. To tackle this issue, this paper introduces a reversible spiking neural node that eliminates the need to store intermediate states during backpropagation by recalculating the states on-the-fly. The authors claim good performance and memory reduction of the proposed SNN node in several vision classification tasks.

**Strengths:**

1. The research topic is both intriguing and important. The substantial training cost associated with SNNs, particularly the considerable memory overhead, has troubled the research community, undermining the potential of SNNs. Therefore, it is imperative to propose various methods to curtail the training cost.

2. The experiments demonstrate a substantial reduction in memory usage.

**Weaknesses:**

1. The description of the neural node in Section 3.2 is convoluted and hard to follow. The underlying logic of the proposed neural node remains unclear. The motivation behind introducing such a neural node is ambiguous. The rationale for segregating the states into two groups at each time step is not motivated. The necessity of introducing $\hat{V^t}$ (Eqs. 3 and 6) is not clearly justified. Moreover, is $[Y_1,Y_2]$ identical to $[X_1,X_2]$ of the next layer? Is $[\hat{V_1^t},\hat{V_2^t}]$ identical to $[V_1^{t+1},V_2^{t+1}]$? To enhance clarity, I recommend that the authors initially expound upon the LIF model, introduce the components of the model that prompt dissatisfaction, and subsequently describe their modifications based on the LIF model.

2. Based on my understanding, $[Y_1, Y_2]$ represents the transmitted signal to other SNN nodes. The issue at hand is that these signals are not binary spikes according to eqs. 2 and 5. However, one of the appealing aspects of SNNs is their utilization of binary events for information processing. In essence, a spiking neuron is active only when it encounters spikes, enabling an event-driven regime and an energy-efficient system. However, the proposed neuron nodes deviate from this advantageous feature by transmitting real-valued signals: neurons are active all the time, and the communication cost is high.

3. Why the neural node can achieve layer reversibility during training? Considering the feedforward connection from the l-th layer to the (l+1)-th layer, $x^{l+1}[t] = W * y^l[t]$, where W is not invertible. I am curious about the approach taken to recalculate $y^l[t]$ from $x^{l+1}[t]$. After all, VGG is used as the network architecture, where the weight matrice are not invertible.

4. While the issue of high training costs is particularly pronounced for large-scale datasets such as ImageNet, this work exclusively tests the proposed method on small-scale datasets (including Tiny-ImageNet). Consequently, the results may not be as convincing, primarily because the training cost for small-scale tasks is relatively affordable. It would be beneficial to extend the evaluation to more computationally demanding datasets to underscore the true efficacy and efficiency of the proposed method.

**Questions:**

See Weaknesses.

---

> ### Author Response · Authors · 2023-11-14
>
> Thank you for your valuable feedback on our work. We would like to address each of the queries you have raised regarding our work in turn.
>
> **Response to *Weakness 1.1 Clarification on spiking neuron Description***:
>
> The rationale behind introducing the reversible spiking neuron lies in enabling the recalculation of activation values during backpropagation. This approach eliminates the need to store activation values during the forward propagation phase, thereby significantly reducing memory usage. To facilitate the recalculation, our model incorporates a symmetric design, wherein states are divided into two groups.
>
> For a more comprehensive understanding of this process, please refer to **Common Question 1** in the **Global Response** section, where we detail how our reversible spiking neuron substantially reduces memory usage and explain the reversibility of our spiking neuron.
>
> **Response to *Weakness 1.2 Symbol classification***:
>
> To address your concern about clarifying the variables in our equations, we have provided a detailed explanation of the symbol system employed in our paper within **Common Question 2** of the **Global Response** section. We used the original LIF model as an example to explicitly delineate the meanings of various symbols in our formulas. This illustration helps to distinguish the symbols used in our work from those in the original LIF models.
>
> Additionally, we would like to address the two specific questions you raised in the reviews:
> - The symbols $[Y_1,Y_2]$ represent the output of each spiking neuron. After passing through intermediate layers, such as CNN or LayerNorm layers, they transform into the input $[X_1,X_2]$ for the next spiking neuron.
> - The notation $[\hat{V_1^{t}},\hat{V_2^{t}}]$ is the same as $[{V_1^{t}},{V_2^{t}}]$. The hat symbol over $V$ was used to denote that the voltage $V$ undergoes in-place modification during the computation process. We have replaced $\hat{V}$ with $V$ for clarity in our revised version.
>
> **Response to *Weakness 2. signal transmission real-valued vs. binary spikes***:
>
> Despite the intrinsic appeal of SNNs due to their use of binary events for information processing, the exploration of real-valued outputs in SNNs is a dynamic area of research. A noteworthy example can be found in the work of Guo et al. [1] (ECCV 2022). Their research demonstrates the feasibility of utilizing real-valued spikes in SNNs while maintaining the advantages of binary spikes.
>
> **Response to *Weakness 3. Explanation for layer reversibility during training***:
>
> Firstly, it is important to highlight that in our model, the transmission of input features $X$ through the spiking neuron does not involve any weight computation.
>
> Secondly, for layers where weights play a role, such as CNN layers, constructing reversible layers is feasible. A notable illustration of this concept is found in the work of Gomez et al. [2] (NIPS 2017). Since weights are updated only after the completion of the gradient calculation, it allows for the precise recalculation of the original activation values using the unchanged, original weights.
>
> To ensure the integrity of our model, we have rigorously verified the consistency between all inputs in the forward operation and the outputs from the inverse operation. This verification was conducted using `torch.allclose(rtol=1e-06, atol=1e-10)`, confirming the accuracy and reliability of our layer reversibility approach during training.
>
> **Response to *Weakness 4. Training Cost and Scalability***:
>
> Thank you for highlighting this important aspect. In response, we have included an additional experiment in the **Common Question 3** of our **Global Response** section. This experiment specifically targets heavy GPU load tasks, with a focus on training on ImageNet-1k using a single GPU with 24GB memory. The results demonstrate that our method successfully avoids out-of-memory issues even at high batch sizes. We believe this additional data will effectively address your concerns regarding the scalability and computational efficiency of our approach in more demanding scenarios.
>
> Thank you once again for your insight reviews! We hope that our responses have adequately addressed and resolved your doubts. Should you have any further concerns or questions at any time, please do not hesitate to let us know. We greatly appreciate your continued guidance and support.
>
> **Reference**
>
> [1] Guo, Y., Zhang, L., Chen, Y., Tong, X., Liu, X., Wang, Y., ... & Ma, Z. (2022, October). Real spike: Learning real-valued spikes for spiking neural networks. In the European Conference on Computer Vision (pp. 52-68). Cham: Springer Nature Switzerland.
>
> [2] Gomez, A. N., Ren, M., Urtasun, R., & Grosse, R. B. (2017). The reversible residual network: Backpropagation without storing activations. Advances in neural information processing systems, 30.

---

> > ### Comment · Reviewer_j4UM · 2023-11-23
> >
> > The rebuttal addresses some of my concerns. Now I am clear about the underlying logic of the proposed neural node. However, the real-valued communication signal still concerns me.
> >
> > First, although Guo et al. [1] proposes to use real-valued 'spikes' during training, the trained SNNs can be recovered to emit binary spikes. Therefore, the obtained models are still standard SNNs that can be implemented on some common neuromorphic hardware, maintaining the event-driven architecture and the good energy efficiency.
> >
> > Second, if the transmitted signals are real-valued in an SNN, such an SNN is not as effective as the corresponding DNN, is not energy-efficient on hardware, and is hard to train. Then it is better to simply use DNNs.

---

### Official Review · Reviewer_JXdn · 2023-10-27

**Soundness:** 3 good
**Presentation:** 2 fair
**Contribution:** 2 fair
**Rating:** 5
**Confidence:** 4

**Summary:**

The paper proposed a framework for zero memory footprint spiking neural network training, which includes a reversible SNN node design and a streamlined backpropagation algorithm. The contributions of the paper are the reduction of memory usage and training time, making SNNs more feasible for resource-limited environments such as IoT-Edge devices.

**Strengths:**

1. This study provides vivid illustrations for the computations of the proposed model, which makes it easy and clear for readers to follow.

2. The experimental results look like convincing.

**Weaknesses:**

Certainly, the authors provide a detailed introduction to spiking computations. However, it is unclear which component or computational step contributes to the memory savings. This lack of clarity might give the impression that the paper reads like a technical report. Providing further elaboration on the modifications and corresponding improvements proposed by this work would be highly beneficial.

In addition, there are a few instances of unclear phrasing:

It is preferable to use "spiking neuron" instead of "spiking neural node" in the machine learning community.

The overall layout of the paper could benefit from improvement, especially on pages 4-6, which can be somewhat arduous to read.

While the derivation of the formula is quite clear, there are still areas that could be refined. For example, using notation like $[V_{11}, V_{21}; V_{12}, V_{22}]$ may be more effective than the current [v_1, v_3; v_2, v_4], which is then divided into V[1] and V[2]. Additionally, the case distinction between V and v may not be of significant importance.

**Questions:**

See weaknesses.

I will consider raising my score if the authors fixed my doubts.

---

> ### Author Response · Authors · 2023-11-14
>
> Thank you for your valuable feedback on our work. We would like to address each of the queries you have raised regarding our work in turn.
>
> **Response to your first concern *it is unclear which component or computational step contributes to the memory savings***:
>
> Our approach eliminates the necessity to store activation values, which traditionally consume substantial memory. Instead, the neuron in our work can recalculate the activation values during the backpropagation phase of SNN training, thereby achieving significant memory efficiency.
>
> For a more comprehensive understanding of this process, please refer to **Common Question 1** in the **Global Response** section, where we detail how our reversible spiking neuron substantially reduces memory usage and explain the reversibility of our spiking neuron.
>
> **Response to your *Improving Terminology Consistency* concern**:
>
> We are now using "spiking neuron" throughout the revised version of our paper to align with common usage in the machine learning community.
>
> **Response to your *Refining Notation and Formula Presentation* concern on pages 4-6**:
>
> We have revised the notation throughout our paper, as evident in the updated **Figure 3**. In the refined approach, uppercase variables represent matrices, while lowercase variables denote elements within those matrices. For instance, $V$ denotes the input voltage matrix, and $𝒗_{11}$ refers to the first element of matrix $V$.
>
> Additionally, we have replaced $\hat{V}$ with $V$ for clarity. Previously, the hat symbol above $V$ was intended to indicate that $V$ is an in-place changing variable.
>
> Thank you once again for your valuable insights and feedback. We hope that our responses have adequately addressed and resolved your doubts. Should you have any further concerns or questions at any time, please do not hesitate to let us know. We greatly appreciate your continued guidance and support.

---

> > ### Comment · Reviewer_JXdn · 2023-11-14
> > **Responses**
> >
> > After reviewing the rebuttals and comments, several uncertainties have been addressed. However, it remains unclear which component or computational step is responsible for the observed memory savings. To be more precise, how efficient is the inverse function saving?
> >
> > I've noticed that other reviewers have posed similar questions. It is hoped that the authors can provide a more explicit response to this inquiry.

---

> > > ### Author Response · Authors · 2023-11-14
> > >
> > > Thanks for your reply!
> > >
> > > The memory saved by our method comes from all the activation values other than the potential voltage in the original SNN. In the example of VGG-13 architecture with ten timesteps,  as shown in our **Figure 2**,  our SNN saves 90.9% activation value memory compared to the original SNN.
> > >
> > > In SNN training, the main memory consumption comes from activation values, which include the spikes output at each layer and a state (the voltage potential $V$, similar to the hidden state in RNNs). Due to the reversibility of our spiking neurons, we do not need to store any output spikes and instead recalculate these values during back propagation.
> > >
> > > The overall computation process is as follows: after passing through each spiking neuron, input features $X$ are computed into output spikes $Y$. While the original SNN stores both $X$ and $Y$ in memory, our approach clears the input features  $X$, which is also the output of previous layers, from memory after each step. This is achieved through a simple line of code: `inputs[0].storage().resize_(0)`. **The inverse function itself does not reduce memory usage. Its role is to assist us in recalculating the activation values that would otherwise need to be stored, in order to perform gradient updates during the backpropagation process.**
> > >
> > > Thank you for your patience. Please let us know if you have any questions or concerns.

---

### Official Review · Reviewer_u1jx · 2023-10-30

**Soundness:** 3 good
**Presentation:** 3 good
**Contribution:** 3 good
**Rating:** 8
**Confidence:** 5

**Summary:**

This paper proposes a new reversible spiking module to circumvent the high memory cost of storing the spatial-temporal computational graph needed for (surrogate) gradient-based training spiking neural networks. Their method trades memory for additional computation during the backward pass, but they introduce a more efficient way to compute the gradient of the reversible module compared to how it is done usually in revertible neural networks. They demonstrate quantitative improvements in terms of memory and compute time of the gradient compared to many previous SNNs approaches, on various vision tasks (CIFAR10/100, DVCGesture, Tiny-ImgaNet...) and network architectures, while maintaining the same level of accuracy.

**Strengths:**

**Originality:** The paper is original because it combines research from different fields: invertible neural networks and spiking neural networks. To my knowledge it has never been done before.

**Quality:** The quality is good, the paper is very detailed, very quantitative and thorough in its comparison with other work.

**Clarity:** The paper is clearly written, though some sentences could be better formulated.

**Significance:** The approach is significant since compute is generally cheaper than memory access in hardware, so to me it makes sense to trade one for the other in the context of edge AI, which is the main use case of SNNs.

**Weaknesses:**

The paper is already good, but one weakness I see is that the approach is only tested on static vision benchmarks that arguably do not require a lot of temporal processing.

The paper would become great if the RevSNN approach was tested on benchmarks that require long sequences, and where the other approaches fail because of out-of-memory. An example of such a task could be image classification from sequences of pixels, as described in [1], but other tasks are possible. The goal is to find a setting where only RevSNN manages to learn the task thanks to its low memory requirement.

[1] Tay, Yi, et al. "Long range arena: A benchmark for efficient transformers." arXiv preprint arXiv:2011.04006 (2020).

**Questions:**

Questions:

- Does the reversible nature of the RevSNN imposes any constraint on the dimensionality of the layers? I am thinking that the dimensions have to be the same for the module to be invertible. If yes, how is it handled in architectures like VGG?

- I don't understand why Figure 4 mentions GNN (Graph Neural Network?), whereas the architectures in the tables are all ConvNets?

- It would be nice if the plots in Figure 6 shared the same y-axis, so that comparison is easier when varying the timesteps.

The authors might want to consider citing [2] and [3], as I think they are related to their approach.

[2] Bauer, Felix C., et al. "EXODUS: Stable and efficient training of spiking neural networks." Frontiers in Neuroscience 17 (2023): 1110444.
[3] Gomez, Aidan N., et al. "The reversible residual network: Backpropagation without storing activations." Advances in neural information processing systems 30 (2017).

---

> ### Author Response · Authors · 2023-11-14
>
> Thank you very much for your recognition of our work! Here are our responses to your questions:
>
> **Response to *Weakness1***:
>
> We agree that evaluating our approach on a mix of both static and dynamic benchmarks is crucial for a comprehensive assessment.  To this end, in the paper we discussed the comparison of our work with the SOTA methods on two static vision datasets, CIFAR10 and CIFAR100, as well as on two dynamic vision datasets, **DVS-CIFAR10** and **DVS128-Gesture**.  These comparative results are comprehensively detailed in **Table 1** of our paper, demonstrating the effectiveness of our approach on varied types of datasets.
>
> **Response to *Weakness 2***:
>
> Thank you for highlighting this important aspect. In response, we have included an additional experiment in the **Common Question 3** of our **Global Response** section. This experiment specifically targets heavy GPU load tasks, with a focus on training on ImageNet-1k using a single GPU with 24GB memory. The results demonstrate that our method successfully avoids out-of-memory issues even at high batch sizes. We believe this additional data will effectively address your concerns regarding the scalability and computational efficiency of our approach in more demanding scenarios.
>
> **Response to *Question 1***:
>
> RevSNN imposes no constraints on the dimensionality of the layers.  However,  the dimension of a specific layer does have to be the same for the layer to be invertible.
>
> In a SNN adapted from conventional VGG architectures, a spiking neuron is typically  added after the CNN layers at the construction stage. In pseudocode,  this step is to modify `layer += [nn.Conv2d(in_channels, out_channels]`  to `layer += [nn.Conv2d(in_channels, out_channels], neuron.LIFNode()]`. The reversibility of the two type of layers are discussed below:
> * Spiking neuron: Reversibility is inherently ensured due to its identical input and output dimensions
> * CNN layers:
> For CNN layers with identical input and output dimensions, we can apply the techniques from your suggested reference, *The Reversible Residual Network: Backpropagation Without Storing Activations*, to facilitate reversibility.
> For CNN layers with different  input and output dimensions, reversibility cannot be achieved due to the inherent discrepancy in dimensions. However, CNN layers in most VGG models have identical input and output dimensions, leaving this issue with minor affect to the overall memory cost.
>
> **Response to *Question 2***:
>
> Thanks a lot for pointing out our typo. We have carefully reviewed and corrected this typo in the revised version of our paper. We hope this amendment addresses your concern and enhances the clarity and accuracy of our work.
>
> **Response to *Question 3***:
>
> Thank you for your valuable suggestion. To facilitate easier comparison, we have updated **Figure 6** so that all plots now share the same y-axis. We hope this alignment can enhance clarity and improve the interpretability of the comparisons.
>
> **Response to *Citation Suggestion***:
>
> We appreciate your valuable suggestion. In line with your recommendation, we have included citations to the two works you suggested in our revised paper.
>
> Thank you once again for your recognition of our work! We hope that our responses have adequately addressed and resolved your doubts. Should you have any further concerns or questions at any time, please do not hesitate to let us know. We greatly appreciate your continued guidance and support.

---

> > ### Comment · Reviewer_u1jx · 2023-11-22
> >
> > Thanks for answering my points and providing additional experiments, I increased my confidence score accordingly.

---

> > > ### Author Response · Authors · 2023-11-22
> > > **Thank you!**
> > >
> > > Thank you for your recognition of our work!  We appreciate your guidance and will update our final version accordingly.

---

### Official Review · Reviewer_Umsy · 2023-11-02

**Soundness:** 3 good
**Presentation:** 1 poor
**Contribution:** 3 good
**Rating:** 5
**Confidence:** 4

**Summary:**

This paper addresses the challenge of memory consumption in the training of Spiking Neural Networks (SNNs). The authors design a special forward and backward computation pattern for SNNs and it does not need to store the intermediate features. Experiments were conducted to show the GPU memory efficiency.

**Strengths:**

+ The paper studies how to reduce the memory footprint of the SNNs during training, which could be an important issue.

+ Though the presentation is not clear, the proposed framework seems to be novel.

**Weaknesses:**

- The methodology part of this work is not well-presented. There are no intuitions or motivations to demonstrate why we have to design this symmetric forward/backward implementation. The notation system is very chaotic, the authors did not mention what is the input/output/membrane potential, and how you denote traditional LIF nodes' implementation. The figures are just copies of the equation.

- This work did not reach the same level of accuracy as the recent SOTA SNN works. For example: TEBN [1], MPBN [2].

- Compared to the original SNNs, this new framework increases the training latency. It is okay to have a little bit higher latency, since most memory efficiency methods trade space for time. However, in that case, I hope the authors can demonstrate the real challenging case, i.e. heavy GPU load tasks, like ImageNet-1k training with a single GPU. How much batch size can this method and the traditional method reach and what are the accuracies? These results are missing.


[1] Duan et al., Temporal Effective Batch Normalization in Spiking Neural Networks.
[2] Guo et al., Membrane Potential Batch Normalization for Spiking Neural Networks.

**Questions:**

See my weakness above.

---

> ### Author Response · Authors · 2023-11-14
>
> Thank you for your valuable suggestions and the recommended references. We sincerely appreciate your time in evaluating our work. Our point-to-point responses to your comments are given below.
>
> **Response to *Weakness 1.1: Clarification on Spiking Neural Description***:
>
> The rationale behind introducing the reversible spiking neuron lies in enabling the recalculation of activation values during backpropagation. This approach eliminates the need to store activation values during the forward propagation phase, thereby significantly reducing memory usage. To facilitate the recalculation, our model incorporates a symmetric design, wherein states are divided into two groups.
>
> For a more comprehensive understanding of this process, please refer to **Common Question 1** in the **Global Response** section, where we detail how our reversible spiking neuron substantially reduces memory usage and explain the reversibility of our spiking neuron.
>
> **Response to *Weakness 1.2: Symbol classification***:
>
> To address your concern about clarifying the variables in our equations, we have provided a detailed explanation of the symbol system employed in our paper within **Common Question 2** of the **Global Response** section. We used the original LIF model as an example to explicitly delineate the meanings of various symbols in our formulas. This illustration helps to distinguish the symbols used in our work from those in the original LIF models.
>
> To directly respond to your specific query regarding the definitions of input, output, and membrane potential:
> 1. **Input potential** at time $t$ is denoted by $V^{t-1}$, representing  the neuron's potential at the previous time step.
> 2. **Output potential** at time $t$  is denoted by $V^{t}$, indicating the neuron's potential at the current time step.
> 3. **Membrane potential** at time $t$ is denoted as $M^{t}$, bridging the neuron's states between the previous and the current time step.
>
> **Response to *Weakness 2***:
>
> We appreciate your comparison with recent SOTA SNN works. Contrary to the concern raised, our results not only achieve higher accuracy than TEBN but also match the accuracy levels of MPBN.
>
> In our research, we have incorporated our reversible spiking neuron into several existing SOTA SNN methodologies, as detailed in **Table 2**. This was done while maintaining other aspects of their setups unchanged, allowing for a direct comparison of the impact of our innovation on accuracy. Our RevDSR model achieves an accuracy of $95.35$% on CIFAR-10 and $78.21$% on CIFAR-100. Upon reviewing TEBN and MPBN, we found that TEBN attains a maximum accuracy of $94.71$% on CIFAR-10 and $76.41$% on CIFAR-100. Meanwhile, MPBN shows a variable accuracy range from $92.22$% to $96.47$% on CIFAR-10 and $70.79$% to $79.51$% on CIFAR-100 under different settings.
>
> **Response to *Weakness 3***:
>
>  Thank you for highlighting this important aspect. In response, we have included an additional experiment in the **Common Question 3** of our **Global Response** section. This experiment specifically targets heavy GPU load tasks, with a focus on training on ImageNet-1k using a single GPU with 24GB memory. The results demonstrate that our method successfully avoids out-of-memory issues even at high batch sizes. We believe this additional data will effectively address your concerns regarding the scalability and computational efficiency of our approach in more demanding scenarios.
>
> Thank you once again for your valuable insights and feedback. We hope that our responses have adequately addressed and resolved your doubts. Should you have any further concerns or questions at any time, please do not hesitate to let us know. We greatly appreciate your continued guidance and support.

---

> > ### Comment · Reviewer_Umsy · 2023-11-20
> > **Reply**
> >
> > Thanks for your response. Part of my concerns have been addressed. However, some key weakness still remains.
> >
> > - Lack of intuition and motivation. Even though the authors provided the original LIF formula, it is unclear what aspects of the original LIF or existing work make GPU training heavy, which part of LIF are you trying to solve, what motivates you to design this reversible neuron. The reviewer expects to see a detailed motivation in Section 3.1. Currently, it is just an example showing memory gains and nothing more.
> >
> > - Compared to SOTA work, RevDSR reaches the same accuracy level but not under the same #timesteps. What is the accuracy of RevDSR using T=2? Is it the same with TEBN? What is the neural architecture, #timesteps in ImageNet experiments and does it have similar accuracy with TEBN, too?

---

> > > ### Author Response · Authors · 2023-11-21
> > >
> > > Thank you for your feedback! We appreciate the opportunity to clarify our approach.
> > >
> > > ***For Q1:***
> > >
> > > SNN models, like the LIF model, require retaining activation values for **each timestep across all layers** of the model, leading to substantial memory usage. Our work aims to eliminate the need to store **any activation values ($X^t$ and $Y^t$) at each time step during training.** To aid in understanding, we reiterate the explanation of the LIF model:
> > >
> > > >1. *Charge step*: The calculation of membrane potential at each time step involves storing the previous voltage and input features, leading to a high memory footprint.
> > >
> > > >$M^t = V^{t-1} + \frac{1}{\tau}\cdot(X^t - V^{t-1})$
> > >
> > > >Here, $V^{t-1}$ represents the neuron’s voltage at time step $t - 1$, and $X^t$ denotes the input features at time step $t$. The term $\frac{1}{\tau}$ acts as a scaling factor. These variables are used to calculate the membrane potential $M^t$ of the spiking neuron at time step $t$.
> > >
> > > >2. *Fire step*: The determination of the neuron's output based on the membrane potential and threshold voltage further adds to the memory usage.
> > >
> > > >$Y^t = \mathcal{H}(M^t - V_{th})$
> > >
> > > >This equation calculates the output at time $t$, denoted as $Y^t$, using the Heaviside function $\mathcal{H}$. It is applied to the difference between the membrane potential $M^t$ and a predefined constant threshold voltage $V_{th}$.
> > >
> > > >3. *Reset step*: Updating the voltage for the next time step requires memory to store the current output and membrane potential.
> > >
> > > >$V^{t} = (1 - Y^t) \odot M^{t} + Y^t \cdot V_{res}$
> > >
> > > >The updated voltage $V^{t}$ is determined by the output $Y^t$ and a constant reset voltage $V_{res}$.
> > >
> > > >In this context of the entire process, $V_{t -1}$ is the input voltage, $V_{t}$ is the output voltage and $M^{t}$ is the membrane potential.
> > >
> > > Our research is driven by the goal of minimizing this memory footprint. The proposed reversible neuron functions as a layer in the original SNN architecture, with input $X^t$ and output $Y^t$. However, all the $X^t$，$Y^t$ values can be recalculated during the backpropagation process, obviating the need for their storage. While previous works, such as Skipper [1], have utilized checkpoint techniques to reduce memory consumption by storing checkpoints instead of activation values, our approach takes a further step.  It reduces the memory footprint by eliminating the need to save checkpoints, achieving a near-zero memory footprint design for SNNs.
> > >
> > > ***For Q2:***
> > >
> > > Our focus is on enhancing the memory efficiency in SNN models. Typically, SNN models with a higher number of time steps tend to consume more memory. With this prerequisite, we picked **DSR** with T=20 (same as the original paper) and **replaced the original spiking neuron with our reversible spiking neuron** to obtain our **RevDSR**. This experiment was designed to show performance retention and memory enhancement with the reversible spiking neuron.
> > >
> > > Due to time constraints during rebuttal, we haven’t integrate our reversible spiking neuron with TEBN. We recognize the importance of a comprehensive evaluation across various SOTA models. In our final version, we plan to include TEBN, following a similar process as with DSR, i.e., replacing its spiking neuron with our reversible spiking neuron. This will provide a broader understanding of the effectiveness of our approach in various SNN architectures.
> > >
> > > To address your specific queries:
> > >
> > > 1. The performance of ReDSR on CIFAR10 with Timstep=2 is  $94.20$%, while the accuracy of TEBN is $94.57$%.
> > >
> > > 2. The following table provides detailed information about the neural architectures, time steps, and accuracies in the ImageNet experiments. Our RevSNN model achieves an accuracy of $63.78$%, while TEBN reports an accuracy of  $64.29$%, both with a time step of 4.
> > >
> > > |   Method |Architecture | Timestep |   Accuracy(%) | Batch Size 128 | Batch Size 256 | Batch Size 512 |
> > > |:------------:|:-----------:|:-----------:|:-----------:|:----------------:|:----------------:|:----------------:|
> > > | Hybrid Training [2]|ResNet34|250| $61.48%$ | Out-of-Memory | Out-of-Memory | Out-of-Memory |
> > > | Tandem Learning [3]|AlexNet|10|  $50.22%$ | $✓$ | Out-of-Memory | Out-of-Memory |
> > > | OTTT [4]|NF-ResNet-34|6|  $64.16%$ | $✓$ | $✓$ | Out-of-Memory |
> > > | **RevSNN(Ours)** |ResNet-18 |4 | $63.78%$ | $✓$ | $✓$ | $✓$ |
> > >
> > > Should this response adequately address your concern and clarify our motivation, we will proceed to update the **Sec. 3.1** and extend experiments. We welcome any further questions and are always open to feedback. Thank you once again for your constructive suggestions.
> > >
> > > **Reference**
> > >
> > > [1] Skipper: Enabling efficient SNN training through activation-checkpointing and time-skipping
> > >
> > > [2] Enabling Deep Spiking Neural Networks with Hybrid Conversion and Spike Timing Dependent Backpropagation
> > >
> > > [3] A Tandem Learning Rule for Effective Training and Rapid Inference of Deep Spiking Neural Networks
> > >
> > > [4] Online Training Through Time for Spiking Neural Networks

---

### Author Response · Authors · 2023-11-14
**Global Response**

We would like to express our sincere gratitude to all the reviewers for the time, effort, and expertise you have invested in reviewing our paper. In this global response, we aim to comprehensively address the common questions and concerns.

**Enhancements and Modifications in the Revised Paper**:

We have updated our paper following the suggestions of the reviewers:
1. Revision of notations in **Figure 3**.
2. Replacing $\hat{V}$ with $V$ in **Section 3** .
3. Correction of Typo in **Figure 4**.
4. Aligning y-axis in **Figure 6** for easier comparison.
5. Including citations to works suggested by the reviewer.
6. Replacing "spiking neural node“ by "spiking neuron".

**Common Question 1 *Motivation and Implementation of the reversible Framework design***:

1. *Why can the reversible framework save memory?*

 Central to our design is the reversible framework's ability to minimize memory consumption during SNN training. Traditionally, substantial memory is allocated to activation values necessary for gradient computation during backpropagation. Our approach innovatively circumvents this by not retaining any activation values during forward propagation. Instead, we introduce an inverse function that recalculates these values precisely when needed for gradient computation.

2. *How to implement reversibility in Spiking Neurons?*

 Our paper details the computational formulas crucial for implementing the reversible approach in **Section 3**.

**Forward computation process (Equations 1-6)**  demonstrates how to two inputs: the input features $X$ and the potential of the spiking neuron at the previous time step$V^{t-1}$ to generate two outputs: the output features $Y$ and the potential of the spiking neuron at the current time step $V^{t}$.

**Inverse computation process (Equations 7-10)** shows how to use the output features $Y$ and the potential of the spiking neuron at the current time step $V^{t}$ to recalculate the input features $X$  and the potential of the spiking neuron at the previous time step $V^{t-1}$.

Dividing states into two groups evenly builds sufficient relationships, thus allows the inverse process to reconstruct the original input values.

**Common Question 2 *Comparing our formula with the LIF model and clarifying symbol meanings***:

The original LIF model comprises three key steps:

1. *Charge step*

$M^t = V^{t-1} + \frac{1}{\tau}\cdot(X^t - V^{t-1})$

Here, $V^{t-1}$ represents the neuron’s voltage at time step $t - 1$, and $X^t$ denotes the input features at time step $t$. The term $\frac{1}{\tau}$ acts as a scaling factor. These variables are used to calculate the membrane potential $M^t$ of the spiking neuron at time step $t$.

2. *Fire step*

$Y^t = \mathcal{H}(M^t - V_{th})$

This equation calculates the output at time $t$, denoted as $Y^t$, using the Heaviside function $\mathcal{H}$. It is applied to the difference between the membrane potential $M^t$ and a predefined constant threshold voltage $V_{th}$.

3. *Reset step*

$V^{t} = (1 - Y^t) \odot M^{t} + Y^t \cdot V_{res}$

The updated voltage $V^{t}$ is determined by the output $Y^t$ and a constant reset voltage $V_{res}$.

In this context of the entire process, $V^{t -1}$ is the input voltage, $V^{t}$ is the output voltage and $M^{t}$ is the membrane potential.

In our paper, most symbols carry the same significance as presented above. However, all inputs and outputs in our work are divided and processed in two separate parts, indicated by subscript $1$ or $2$. Here, ‘$1$’ refers to the first part of the variable after division, and ‘$2$’ to the second.

**Common Question 3 *Experiment involving a heavy GPU load task***:

To test our design’s performance under heavy duty GPU load, we have conducted experiments using ImageNet on an RTX6000 GPU (24GB RAM), comparing it against SOTA memory-efficient approaches. The results, shown in the table below, confirm our design's robust handling of high batch sizes (up to 512) without memory issues, demonstrating its effectiveness in resource-intensive scenarios.

|   Method |   Accuracy(%) | Batch Size = 128 | Batch Size = 256 | Batch Size = 512 |
|:------------:|:-----------:|:----------------:|:----------------:|:----------------:|
| Hybrid Training [1]| $61.48%$ | Out-of-Memory | Out-of-Memory | Out-of-Memory |
| Tandem Learning [2]|  $50.22%$ | $✓$ | Out-of-Memory | Out-of-Memory |
| OTTT [3]|  $64.16%$ | $✓$ | $✓$ | Out-of-Memory |
| **RevSNN(Ours)** | $63.78%$ | $✓$ | $✓$ | $✓$ |


Thank you once again to all the reviewers! We hope our responses address your concerns and welcome any further questions. Your ongoing guidance and support are greatly appreciated.

**Reference**

[1] Enabling Deep Spiking Neural Networks with Hybrid Conversion and Spike Timing Dependent Backpropagation (ICLR 2020)

[2] A Tandem Learning Rule for Effective Training and Rapid Inference of Deep Spiking Neural Networks (TNNLS 2021)

[3] Online Training Through Time for Spiking Neural Networks (NIPS 2022)

---

### Author Response · Authors · 2023-11-19
**ICLR #2819 version3 Updates (11/19)**

Minor updates in Figure $3$ and symbol notations for Equation $7$ to $10$ in Section $3$ for clarity and better presentation.

---

### Author Response · Authors · 2023-11-22
**Further Elaboration of the Motivation**

To address the concerns raised by reviewers regarding the motivation, please let us provide some additional clarification to make it more clear.

Generally, the memory requirements during the training of neural networks consist of two main components: **(1) learnable parameters**, such as weights, and **(2) activation values**, which include the output of each layer, that is also the input to the subsequent layer.

Typically, **the memory needed to store activation values is much greater than that needed for learnable parameters**, and it increases with the batch size. This issue is even more pronounced in SNNs because they comprise multiple time steps. Compared to other networks, SNNs must retain the activation values for all layers across all time steps, rather than just for all layers at a single time point.

Storing activation values is essential for computing gradients during the backpropagation phase of training, which are used to update weights. This is why the memory requirements for inference are substantially lower than for training since there is no need to preserve these activation values for weight updates.

**Our design is aimed at eliminating the need to save these activation values in SNNs**. During our backpropagation process, we first recalculate the necessary activation values for the current gradient update using our designed inverse equations, and then proceed to update the gradients. By employing this approach, we can forego the need to save activation values during training, thereby significantly reducing memory requirements.

If there are any further concerns, please feel free to let us know. We look forward to your further guidance!

---

### Meta-Review · Area_Chair_UgMc · 2023-12-05

**Metareview:**

This paper received mixed reviews initially. Three reviewers are borderline negative while one is strongly positive. The raised issues include unclear technical presentation, insufficient experimental evaluation, unclear memory reduction mechanism, and so on. During the rebuttal and discussion phases, the authors address the raised issues to some extent, but not completely. Concerns regarding the memory and efficiency of developed modules still exist, and do not fully convince the reviewer and AC. Overall, the AC has monitored the whole process and reached a consensus among panels that the current submission is not ready for reporting. The authors are suggested to improve the current paper according to comments. Welcome for the next venue.

**Justification For Why Not Higher Score:**

The network design is not completely new.

**Justification For Why Not Lower Score:**

n/a

---

### Decision · Program_Chairs · 2024-01-16

Reject